DATA RELEASE

# X-ray micro-tomographic data of live larvae of the beetle *Cacosceles newmannii*

Philipp Lehmann[1,2,*], Marion Javal[1], Anton Du Plessis[3], Muofhe Tshibalanganda[3] and John S. Terblanche[1]

1 Centre for Invasion Biology, Department of Conservation Ecology and Entomology, Stellenbosch University, Stellenbosch, South Africa
2 Department of Zoology, Stockholm University, Sweden
3 CT Scanner Facility, Central Analytical Facilities, Stellenbosch University, Stellenbosch, South Africa

## ABSTRACT

Quantifying insect respiratory structures and their variation has remained challenging due to their microscopic size. Here we measure insect tracheal volume using X-ray micro-tomography ($\mu$CT) scanning (at 15 $\mu$m resolution) on living, sedated larvae of the cerambycid beetle *Cacosceles newmannii* across a range of body sizes. In this paper we provide the full volumetric data and 3D models for 12 scans, providing novel data on repeatability of imaging analyses and structural tracheal trait differences provided by different image segmentation methods. The volume data is provided here with segmented tracheal regions as 3D models.

**Subjects** Imaging, Biodiversity, Morphology

**Submitted:** 16 February 2021

* Corresponding author at Department of Zoology, Stockholm University, 10691 Stockholm, Sweden. Tel.: +468164089. E-mail: philipp.lehmann@zoologi.su.se

Preprint submitted at https://doi.org/10.31730/osf.io/2urxf

## INTRODUCTION

In insects, oxygen is supplied through air-filled tubes called tracheae [1, 2]. These originate in ventilatory valves embedded in the cuticle, called spiracles, that lead to a branching network of tubes of decreasing diameter that ultimately end in close proximity to the tissues and cells to which they supply oxygen. The smallest tracheae, with a diameter smaller than 2 $\mu$m, are called tracheoles and the major site of oxygen exchange [3, 4]. X-ray micro-tomography ($\mu$CT) is an emerging tool with which to study tracheal networks [6–13]. Using $\mu$CT has the benefit of allowing reconstruction of the intact tracheal tree in its three-dimensional configuration. However, previous studies have typically made use of dead animals, sometimes frozen or histologically fixed, which can have significant impact on details (e.g. tracheal collapse, or systemic fluid filling) of tracheal structures [14]. Here we investigate tracheal oxygen supply using $\mu$CT on living, sedated larvae of the beetle, *Cacosceles newmannii*. We use the method to quantity volume and area of isolated tracheal trees and compare the relationship between these traits and body mass at two time-points. Furthermore, the relevant $\mu$CT setup, scanning, reconstruction, and visualization methods are explained and the acquired datasets are provided, in the form of 3D volume data (image stacks) and models of the segmented tracheal systems studied.

## IMPLEMENTATION

The longhorned beetle *Cacosceles newmannii* (Coleoptera: Cerambycidae) Thomson 1877, is native to South Africa, Mozambique and eSwatini [15]. For this study we used larvae

collected by hand from KwaZulu-Natal sugarcane farms (South Africa, Entumeni district, 28° 55′ S; 31° 19′ E). Samples were transported to Stellenbosch University and maintained individually at 25 °C in a 16L:8D regime, in 30 ml jars containing sterilized peat and *ad libitum* food provision (bits of 10 cm fresh sugarcane stalk).

The μCT scans were performed with parameters optimized as demonstrated previously [16]. In order to obtain high-quality μCT scans we anesthetized larvae with sevoflurane (Sojourn, Piramal, Bethlehem, USA) in accordance with recommendations for insects [17]. We opted for sevoflurane over sedation using $CO_2$ and cooling, the two most commonly used alternatives, for several reasons. Most importantly, both cooling and $CO_2$ sedation have been shown to influence growth and long-term fitness of animals, unlike sevoflurane, which in *D. melanogaster* led to no measurable negative effects on treated animals [17]. Further, cooling the scanning chamber sufficiently is challenging due to space limitations and potential interference between heat-exchangers and electronic equipment in the scanner. Thus, sevoflurane-induced anaesthesia is an exciting alternative that we tested here in a new setting. Larvae were placed in 15 ml Falcon tubes surrounded by cotton, upon which 100–200 μl of sevoflurane was pipetted. Scans were performed when larvae stopped all movement, in general within 5–10 min after the application of the anaesthetic. Before scanning, a wedge of firm closed-cell foam mounting material was inserted along the back of the larva to firmly keep it in place. Both the foam and cotton are of very low density and easy to separate during subsequent segmentation analyses, and an important consideration when designing μCT experiments on live samples. After the scan larvae were immediately returned to holding jars and in each case recovered fully, i.e. started moving, within 60 seconds upon removal from the sevoflurane-containing Falcon tube.

All samples were scanned (in Falcon tubes) in a General Electric VTomex L240 (General Electric Sensing and Inspection Technologies/Phoenix X-ray, Wunstorff, Germany) μCT-scanner at the CT Scanner Facility at Stellenbosch University, South Africa. Settings were 120 kV and 100 μA for X-ray generation, magnification was set to achieve a voxel size of 15.000 μm. The size of the larvae limited the best possible resolution. At 15 μm voxel size, the head of the largest specimen in the batch could be covered with sufficient field of view to cover two segments of the larva. This voxel size was achieved with a source-detector distance of 600 mm and a source-object distance of 45 mm. Three exceptions were the samples numbered 30, 31 and 32 which were recorded at 16.667 μm. These voxel sizes are needed to correctly load the image stack data provided.

Scanning was performed with each image acquired in 250 ms, with averaging of two images at each rotational step position to enhance image quality. A total of 1500 step positions were used in one full rotation of the sample. Importantly, these settings allowed us to perform scans of approximately 18 minutes (which is relatively short for μCT), to minimize the time animals spent under sevoflurane anaesthesia.

Image segmentation was performed in Volume Graphics VGSTUDIO MAX 3.2. The process aimed at obtaining a region of interest of the air space inside single tracheal trees for each specimen. We scanned the anterior part of the specimen, including the first 3 spiracular openings. For tracheal nomenclature and abbreviations we follow [9] and for analyses, we only considered the tracheal tree branching in from the first metathoracic ($ts_2$). Trees on each side of the larva were considered as replicates for the same specimen. Thus we analyzed two trees per individual larva, and took averages from these two trees for the statistical analyses and figures. A surface determination function was applied to identify



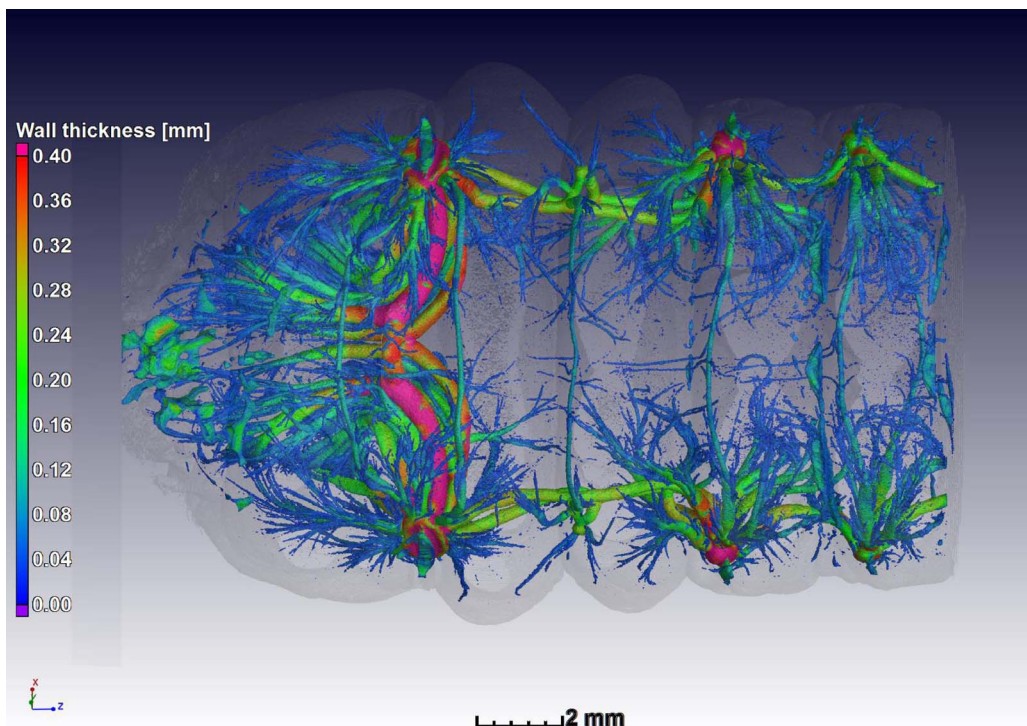

**Figure 1.** Full tracheal system in the scanned region with color coding representing local thickness of the channel, in this case sample nr 5 is shown. See video in the discussion.

the material–air interface—this is equivalent to a global thresholding segmentation but uses a local refinement for sub-voxel edge determination. Following this, the next step was to manually close the tracheal entrances (i.e. all spiracles) using the 3D drawing tool provided by the software. The inner air space of the whole tracheal system was then selected using a region growing tool within this closed area. The next step was to separate the tracheal tree of interest from the rest of the tracheal system, by manually cutting tracheae connecting other spiracles using the drawing tool. In order to standardize the way a tree was isolated from the rest of the system, several landmarks were chosen. First, the dorsal (dlt) and ventral (vlt) longitudinal tracheal trunks on the anterior end of the $ts_2$ spiracle were cut halfway towards the mesothoracic spiracle ($ts_1$) at a characteristic enlargement visible in each specimen. Then, dlt and vlt on the posterior end of the $ts_2$ spiracle were cut just after the atrium. Finally, the dorsal (dc) and ventral (vc) transversal commissures were cut as close as possible to the atrium of the ts2 spiracle. Once a tracheal tree was isolated (i.e. a spiracular opening and all trachea attached to that spiracle), its area and volume were extracted in $mm^2$ and $mm^3$, respectively. Default $3 \times 3 \times 3$ median filtering was employed to de-noise images. Figure 1 is a 3D visualization of the full tracheal system in the scanned region (sample nr 5) near the head, with color coding representing local thickness of the tracheal channel. Figure 2 shows the segmented single tracheal tree, for which STL files are available for all data sets.

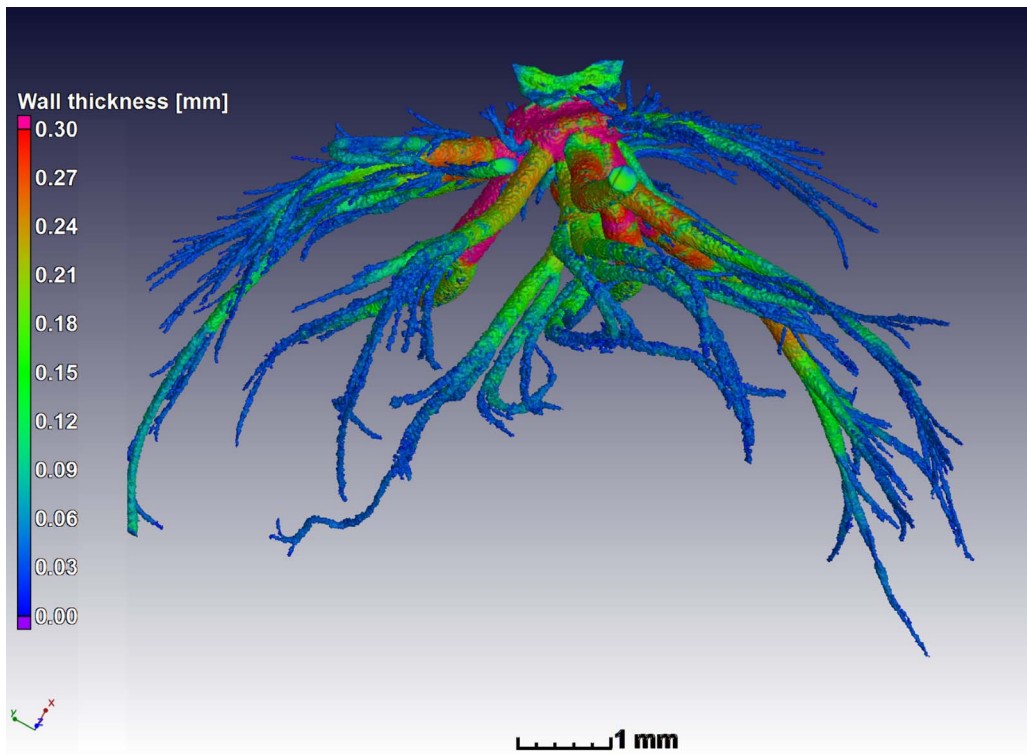

**Figure 2.** Single segmented tracheal tree of sample nr 5 with local color coding for visualization. See video in the discussion.

## AVAILABILITY OF SUPPORTING SOURCE DATA AND REQUIREMENTS

The data presented here are provided in the form of cross sectional image stacks (Tiff files), together this forms the volume data that could be used for further analysis or further segmentation of other features. One segmented tracheal tree from each sample as described above (shown in Figure 2) is provided in the form of a 3D model as STL file, providing guidance to researchers wishing to perform further segmentation of these structures from the volume data. A simplified model of the exterior of the larva in each case is also provided. This simplified model is generated using a down-sampled dataset at 0.1 mm voxel size with subsequent STL generation. In order to demonstrate the potential for usage of higher quality data of the exterior of the samples, one sample (nr 31) is provided with a 500 Mb high quality surface model additionally, showing finer detail of the surface structures of the larva (see Figure 3 for sketchfab versions of the scans).

The data accompany a recent paper [18], where they are combined with data on growth, survival and metabolic rate in the same animals. In that study preliminary analyses suggest matched growth of metabolic demand and oxygen supply over half an order of magnitude of mass gain in the larvae. In fact, the traits suggest overcapacity, rather than capacity deficit, at larger masses.

## DISCUSSION

In the present study we explore the possibility to use μCT based scanning to investigate tracheal growth in live insects. We were successfully able to sedate larvae [17], scan and

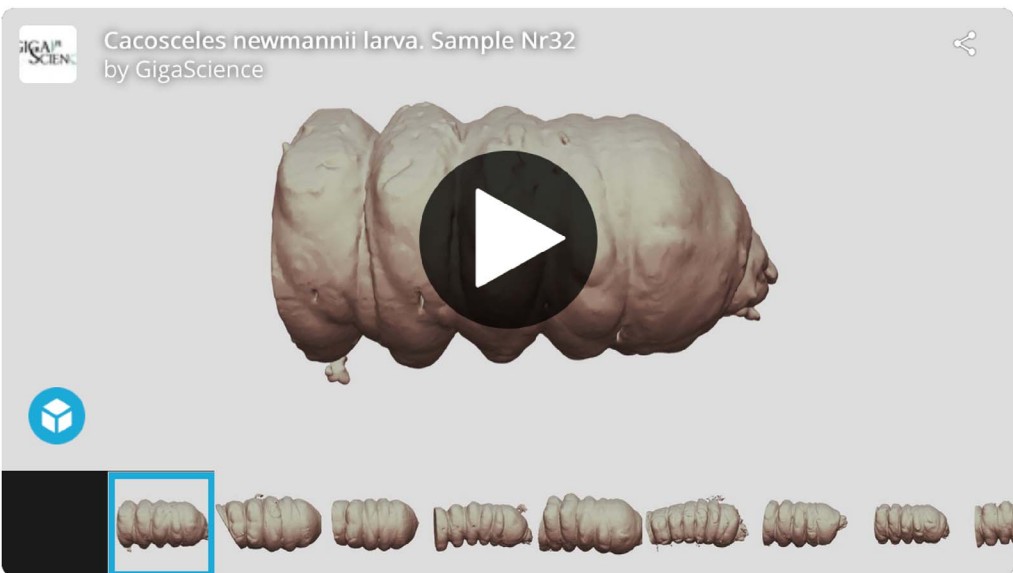

**MicroCT images of Cacosceles newmannii** by **GigaScience** on **Sketchfab**

**Figure 3.** Sketchfab collection of interactive CT images of live larvae of the beetle *Cacosceles newmann.* https://sketchfab.com/GigaDB/collections/microct-images-of-cacosceles-newmannii

reconstruct tracheal structures, and can show that exposure to the anesthetic and μCT radiation dose does not lead to increased mortality or a systematic difference in growth [18]. This opens up new avenues to test morphology, performance and the potential fitness consequences, and thus obtain respiratory anatomical estimates as animals develop, grow and change shape in ways that have not been well explored in insects to date (see the video summary in Figure 4).

One important limitation in the current study was the achieved best resolution of 15 μm. Since the tracheal system consists of tubing with diameters decreasing down to less than 1 μm [4], a potentially large proportion of the tracheal system was not yet quantified. A study of adult *Hypothenemus hampei* beetles shows that the majority of tracheal volume resides in trachea with diameters of less than 10 μm and trachea with a diameter over 15 μm only represent less than 10% of total volume [7]. This could primarily be due to the very small size of these beetles. Indeed, a study on the much larger bodied grasshopper *Schistocerca americana* found a very strong correlation between different traditional methods of quantifying tracheal air volume and a μCT method using voxel sizes of 48 μm [10], indicating that the 15 μm achieved in the present study might be sufficient for the large-bodied larvae used here.

Tracheal volume quantification using μCT are however further complicated by observations in *T. molitor* that varying voxel size (i.e. scan resolution) gives different results depending on the body compartment studied [9, 14]. While using smaller voxel sizes increases total volume estimates on the whole body scale, a less steep relationship is seen if restricting the analysis to only the head and prothorax, probably due to differences in content of tissue with varying metabolic demand [19, 20]. Thus, when using hard cut-offs in tracheal diameter, it is difficult to know how much of total tracheal volume that actually is quantified, and how much is found in trachea and tracheoles with smaller diameters.

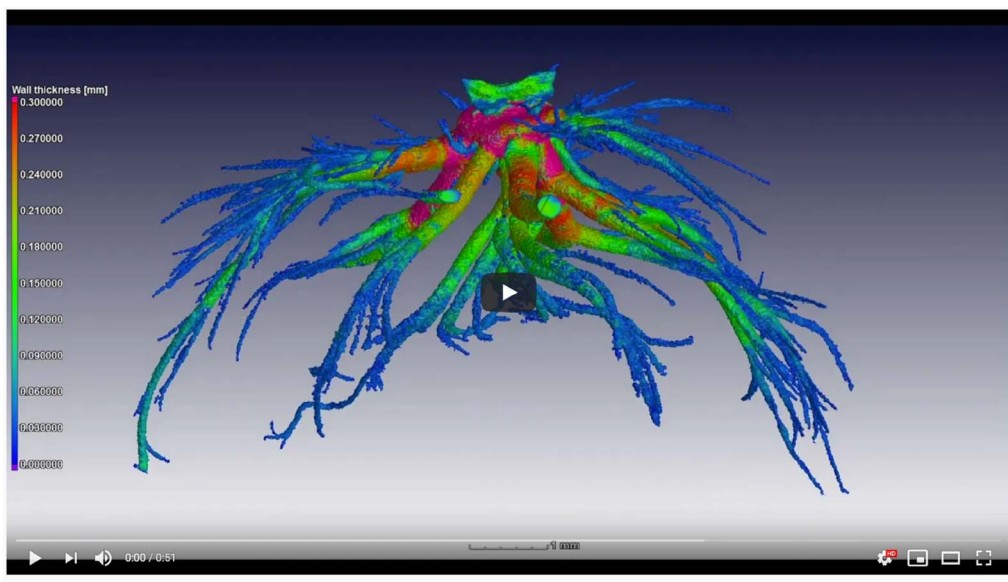

Quantifying insect respiratory structures using microCT data of live larvae of the longhorned beetle

**Figure 4.** Video summary of the work quantifying insect respiratory structures using X-ray micro-tomographic of live larvae of the beetle *Cacosceles newmannii*. See: https://youtu.be/z_WkZzhCpJw

Future studies could explore possibilities to improve minimum resolution, for instance using machine learning algorithms that improve capacity to distinguish tracheal structures in fuzzy images [5]. More generally, it would be important to study insect species across a wide range of body sizes to determine the distribution of tracheal size relationships [6, 21] at different scanning resolutions [14].

## CONCLUSIONS

We investigated the tracheal system in larvae of *C. newmannii*, a large-bodied beetle with pronounced ontogenetic variation in mass. We were able to successfully sedate larvae and perform repeated measurements of tracheal traits down to a resolution of 15 μm on live individuals, opening up new methodological avenues for further study. It seems likely that using living organisms carries the cost of poorer resolution, which needs to be traded off against the benefit of having organs and internal structures in an anatomically appropriate configuration.

## DATA AVAILABILITY

Supporting data is available in the *GigaScience* GigaDB repository [22].

Sketchfab specimen 3D models are available [23] alongside 3D printable models in the Thingiverse repository [24].

## DECLARATIONS
## LIST OF ABBREVIATIONS

$\mu$CT (X-ray micro-tomography), $\mu$m (micrometre), mm (millimetre), cm (centimetre), mg (milligram), g (gram), $CO_2$ (carbon dioxide), $\mu$l (microliter), ml (millilitre), kV (kilovolt), $\mu$A (microampere), ms (millisecond).

## ETHICS APPROVAL AND CONSENT TO PARTICIPATE

Not applicable.

## COMPETING INTERESTS

The authors declare that they have no competing interests.

## AUTHORS' CONTRIBUTIONS

PL, AdP and JST drafted the manuscripts. AdP, MT and MJ generated the data. All authors made comments on the manuscript.

## FUNDING

This work was supported by the Company of Biologists and Journal of Experimental Biology to P.L. (JEB-171103) and through a Center for Invasion Biology (CIB) Fellowship to P.L.d

## ACKNOWLEDGEMENTS

We are thankful for laboratory assistance from Chantelle Smit.

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
