## [Reviewer Report]

Comments on revised manuscriptThe authors have provided new exterior surface reconstructions as requested and additionally the supplementary movies with colour coding of tracheal thickness. The new surface reconstructions are of very high quality and are a great addition to this GigaByte Data Release manuscript.

The authors have addressed all the issues that I raised and it is my recommendation that this manuscript should be accepted for publication in GigaByte.

---

## [Reviewer Report]

Upload additional filesDRR-202102-02/form/Beetle_Figure1.pngReviewer name and names of any other individual's who aided in reviewer Chris ArmitDo you understand and agree to our policy of having open and named reviews, and having your review included with the published papers. (If no, please inform the editor that you cannot review this manuscript.)YesIs the language of sufficient quality?YesPlease add additional comments on language quality to clarify if needed
Are all data available and do they match the descriptions in the paper? YesAdditional CommentsThe authors provide volumetric data, which includes 16-bit TIFF stacks of the anterior part of 12 larvae of the cerambycid beetle Cacosceles newmannii across a range of body sizes. The authors additionally provide 3D surface reconstructions (STL format) of the anterior part of the larval specimen, and of segmented tracheae. The segmentations of the tracheae are of very high quality and interactive 3D models for each of these trachea trees should be made available on GigaDB. In contrast, the segmentations of the anterior part of the specimen are a little bit more problematic. Whereas I see value in these additional 3D reconstructions in that they allow a researcher to understand where in the anterior part of the specimen the tracheal segmentations were extracted, the 3D reconstructions of the anterior part of the beetle larval specimen include debris - which may represent the foam and cotton mounting material that the authors refer to in the manuscript - and additionally a cylinder container which may represent the 15ml Falcon tube in which the specimen was scanned (see attached file Beetle_Figure1.png). This makes it quite difficult to observe the external morphology of the Cacosceles newmannii larval specimens. Of note, the authors state in the manuscript that, “Both the foam and cotton are of very low density and easy to separate during subsequent segmentation analyses, and an important consideration when designing μCT experiments on live samples.” Consequently, I invite the authors to submit improved segmentations of the anterior part of the beetle larval specimens in support of this statement.Are the data and metadata consistent with relevant minimum information or reporting standards? See GigaDB checklists for examples <a href="http://gigadb.org/site/guide" target="_blank">http://gigadb.org/site/guide</a>YesAdditional CommentsI could not find the supplementary movies referred to in the manuscript with the colour coding of tracheal thickness. Please submit the missing movies to GigaByte.Is the data acquisition clear, complete and methodologically sound?YesAdditional CommentsIs there sufficient detail in the methods and data-processing steps to allow reproduction?YesAdditional CommentsIs there sufficient data validation and statistical analyses of data quality? YesAdditional CommentsIs the validation suitable for this type of data?YesAdditional CommentsIs there sufficient information for others to reuse this dataset or integrate it with other data?YesAdditional CommentsAny Additional Overall Comments to the AuthorThe live imaging approach outlined in this GigaByte manuscript is of great interest to the research community and could additionally be used to investigate genotype-phenotype relationships in adult organisms. To illustrate with one example, in the fruitfly Drosophila melanogaster, it is known that Notch-Delta signalling is critical for branching morphogenesis in the tracheal system (Llimargas, 1999, Development 126: 2355-2364) and that this relates to Notch activation at branch points in the larval trachea (Rao et al., 2015, eLife 4:e08666). Whereas it is relatively straightforward to use confocal microscopy to assess tracheal morphology in embryonic and larval Drosophila specimens, it is difficult to morphologically assess tracheal deformations in adult fruitflies with confocal imaging. Consequently, the X-ray micro-tomography live imaging technique outlined in this manuscript highlights a complementary approach that is perfectly suited for investigating tracheal morphology in larger specimens – such as adult fruitflies - and this could be used to explore the relationship between, for example, body mass and tracheal branching in malformed yet viable mutant Drosophila specimens. Consequently, I see great potential for this X-ray micro-tomography live imaging method in understanding phenotypic abnormalities.RecommendationMinor Revision

---

## [Reviewer Report]

Reviewer name and names of any other individual's who aided in reviewer Waisum MaDo you understand and agree to our policy of having open and named reviews, and having your review included with the published papers. (If no, please inform the editor that you cannot review this manuscript.)YesIs the language of sufficient quality?YesPlease add additional comments on language quality to clarify if needed
One typo spotted on line 129: the word 'does' is duplicated.Are all data available and do they match the descriptions in the paper? YesAdditional CommentsAre the data and metadata consistent with relevant minimum information or reporting standards? See GigaDB checklists for examples <a href="http://gigadb.org/site/guide" target="_blank">http://gigadb.org/site/guide</a>YesAdditional CommentsIt would be great if more information on 'Sample storage location' is provided e.g. condition/location of where the live samples are raised.Is the data acquisition clear, complete and methodologically sound?YesAdditional CommentsIs there sufficient detail in the methods and data-processing steps to allow reproduction?YesAdditional CommentsIs there sufficient data validation and statistical analyses of data quality? YesAdditional CommentsIs the validation suitable for this type of data?YesAdditional CommentsIs there sufficient information for others to reuse this dataset or integrate it with other data?YesAdditional CommentsAny Additional Overall Comments to the AuthorFor Figure 1 and 2, I suggest stating which samples are shown in the caption.RecommendationMinor Revision